# Analysis of Parainflammation in Chronic Glaucoma Using Vitreous-OCT Imaging

**DOI:** 10.3390/biomedicines9121792

**Published:** 2021-11-29

**Authors:** María Jesús Rodrigo, Manuel Subías, Alberto Montolío, Silvia Méndez-Martínez, Teresa Martínez-Rincón, Lorena Arias, David García-Herranz, Irene Bravo-Osuna, Julian Garcia-Feijoo, Luis Pablo, José Cegoñino, Rocio Herrero-Vanrell, Ana Carretero, Jesus Ruberte, Elena Garcia-Martin, Amaya Pérez del Palomar

**Affiliations:** 1Department of Ophthalmology, Miguel Servet University Hospital, 50009 Zaragoza, Spain; manusubias@gmail.com (M.S.); oftalmosmm@gmail.com (S.M.-M.); teresamrincon@gmail.com (T.M.-R.); LorAriCam@hotmail.com (L.A.); lpablo@unizar.es (L.P.); egmvivax@yahoo.com (E.G.-M.); 2Miguel Servet Ophthalmology Research Group (GIMSO), Aragon Health Research Institute (IIS Aragon), 50009 Zaragoza, Spain; 3National Ocular Pathology Network (OFTARED), Carlos III Health Institute, 28040 Madrid, Spain; rociohv@farm.ucm.es; 4Biomaterials Group, Aragon Engineering Research Institute (I3A), University of Zaragoza, 50018 Zaragoza, Spain; amontolio@unizar.es (A.M.); jcegoni@unizar.es (J.C.); amaya@unizar.es (A.P.d.P.); 5Department of Mechanical Engineering, University of Zaragoza, 50018 Zaragoza, Spain; 6Innovation, Therapy and Pharmaceutical Development in Ophthalmology (InnOftal) Research Group, UCM 920415, Department of Pharmaceutics and Food Technology, Faculty of Pharmacy, Complutense University of Madrid (UCM), 28040 Madrid, Spain; davgar07@ucm.es; 7Health Research Institute of the San Carlos Clinical Hospital (IdISSC), 28040 Madrid, Spain; 8University Institute of Industrial Pharmacy (IUFI), School of Pharmacy, Complutense University of Madrid, 28040 Madrid, Spain; ibravo@ucm.es; 9Department of Ophthalmology, San Carlos Clinical Hospital, UCM, 28040 Madrid, Spain; jgarciafeijoo@hotmail.com; 10Centre for Animal Biotechnology and Gene Therapy (CBATEG), Universitat Autònoma de Barcelona, 08193 Bellaterra, Spain; ana.carretero@uab.cat (A.C.); jesus.ruberte@uab.cat (J.R.); 11CIBER for Diabetes and Associated Metabolic Diseases (CIBERDEM), 28029 Madrid, Spain; 12Department of Animal Health and Anatomy, School of Veterinary Medicine, Universitat Autònoma de Barcelona, 08193 Bellaterra, Spain

**Keywords:** optical coherence tomography, biomedical imaging, image analysis and processing, vitreous body, glaucoma, animal models, inflammation

## Abstract

Glaucoma causes blindness due to the progressive death of retinal ganglion cells. The immune response chronically and subclinically mediates a homeostatic role. In current clinical practice, it is impossible to analyse neuroinflammation non-invasively. However, analysis of vitreous images using optical coherence tomography detects the immune response as hyperreflective opacities. This study monitors vitreous parainflammation in two animal models of glaucoma, comparing both healthy controls and sexes over six months. Computational analysis characterizes in vivo the hyperreflective opacities, identified histologically as hyalocyte-like Iba-1+ (microglial marker) cells. Glaucomatous eyes showed greater intensity and number of vitreous opacities as well as dynamic fluctuations in the percentage of activated cells (50–250 microns^2^) vs. non-activated cells (10–50 microns^2^), isolated cells (10 microns^2^) and complexes (>250 microns^2^). Smaller opacities (isolated cells) showed the highest mean intensity (intracellular machinery), were the most rounded at earlier stages (recruitment) and showed the greatest change in orientation (motility). Study of vitreous parainflammation could be a biomarker of glaucoma onset and progression.

## 1. Introduction

Chronic glaucoma is the leading cause of irreversible blindness in the developed world [1]. Increased intraocular pressure (IOP) is the modifiable risk factor most strongly associated with the onset and progression of the disease [2]. This pathology causes progressive retinal ganglion cell (RGC) death, resulting in irreversible visual field impairment [3,4]. Campimetric alteration is detected when 25–30% of RGCs have died, leading to late diagnosis. It is therefore necessary to develop new tools and markers for early diagnosis. With the aim of better understanding the pathogenesis of glaucoma, different models of ocular hypertension have been developed in animals, either by genetic engineering [5,6], or by injection of neurotoxic substances [7,8], or by increasing IOP using pre- and post-trabecular mechanisms [9,10,11]. At the same time, there is also a known progression in glaucoma of neurodegeneration not dependent on IOP, which has been related to cellular and humoral immune-mediated response [12,13,14,15].

The eye has traditionally been considered an immunoprivileged organ. This condition allows a transparent optical visual axis to be maintained by preventing neural destruction followed by repair, scarring and subsequent irreversible visual loss in the event of noxa [16]. It is a complex balance of regional immune system specializations based on a baseline immunoinhibitory microenvironment. Multiple components act by regulating the function and activation of innate (monocyte-macrophage) and adaptive cells (cellular and humoral response) [17,18,19]. In pathological situations such as non-infectious uveitis, the eye is subject to acute adaptive inflammatory processes. However, parainflammatory processes characterized by slow and insidious inflammation, mainly with the involvement of innate immune elements, have also been recognized in processes previously thought to be purely degenerative, such as age-related macular degeneration, diabetic retinopathy and glaucoma [20,21,22]. In glaucoma, several signs suggestive of inflammation have been described, such as more acidic pH with increased immune proteins in the aqueous humour and vitreous, as well as activation of astrocytes and microglia and deposits of immunocomplexes and immunoglobulins in the retina identified by histological techniques [23,24,25,26,27,28,29,30]. Microglia are cells of innate immunity derived from blood-borne monocyte-macrophages that migrate during development to the central nervous system and retina where effector functions are acquired to maintain serosal homeostasis and immune surveillance. In the effector organs, these cells present different morphologies depending on their state of activation [28,31,32]. Soma size, assessed using automated in vivo fluorescence image analysis, has been proposed as a significant morphological marker of microglial activation in the neocortex [31]—and subsequently corroborated in glaucomatous mouse retinas—and a positive linear correlation was found with Iba1 staining (specific for microglia) [33]. Microglial activation appears to be the earliest detectable change in the retina [34] that strongly correlates with and predicts the severity of neurodegeneration following acute and chronic damage in glaucoma [35].

The vitreous body is a transparent three-dimensional structure that participates in the cushioning and transmission of light and plays an important role in the metabolism of the different ocular structures under physiological and pathological conditions [36,37,38]. Very few studies have delved into analysis of the vitreous in entities with parainflammation [39]. The vitreous body is composed mainly of water, collagen fibres, hyaluronic acid and hyalocytes. Hyalocytes are hematopoietic cells of monocyte/macrophage lineage (as microglia) that are located at the periphery of the vitreous cavity, mainly at the anterior base of the vitreous next to the ciliary body and in the vicinity of the optic nerve head [40]. Their functions include synthesis of extracellular matrices, participation in immune regulation of vitreous cavity-associated immune deviation (as inhibitors) through phagocytic activity and contractile properties, and modulation of intraocular inflammation with surface receptors for immunoglobulins and complements. Hyalocytes are highly sensitive to local and systemic physiological perturbations. In response to noxa, they are replaced by reproduction, increasing their mitotic activity, and their changes are postulated as early indicators of value in ocular disease [41]. Imaging-based study of changes in the vitreous under conditions of parainflammation could provide a better understanding of the pathophysiology of these diseases.

Most glaucoma studies have focused on neuroretinal analysis (using either histological or imaging techniques) and very few have focused on vitreous analysis, possibly due to the risk derived from the vitreous sampling process [42,43]. Histological studies are time-consuming, susceptible to human bias and not conducive to analysis of large databases. In contrast, optical coherence tomography (OCT) is an objective, fast and cost-efficient technological tool that allows in vivo acquisition of high-resolution cross-sectional images in the order of microns. Latest-generation OCT systems allow users to study the vitreous in a non-invasive manner under normal conditions and in acute and chronic inflammatory processes, it also being possible to evaluate the changes that occur after treatment [44,45,46]. Vitreous opacities analysed using OCT can provide rapid correlation with vitreous and intraretinal ocular inflammation and do not require any correction factor for vitreous histological correlation [47,48]. Recently, our group described for the first time an increase in vitreous signal intensity using OCT after induction of ocular hypertension in an animal model of chronic glaucoma [49]. Our group hypothesizes that, thanks to the optical transparency of the vitreous, the activation and/or increase in the number of hyperreflective vitreous opacities detected in OCT images corresponding to immune cells can be identified more easily than intraretinal ones. Moreover, unlike other techniques, it does not require modification of the animal’s genetics, nor does it require contrast injection to express fluorescence and, therefore, it has greater potential for application in clinical settings. To our knowledge, OCT analysis of the vitreous has not previously been studied in situations of parainflammation in glaucoma. This paper analyses the changes in vitreous signal in rat eyes, measured using OCT and differentiated by sex, in two models of chronic glaucoma induced by ocular hypertension over 6 months. Computational analysis characterizes the hyperreflective opacities at the vitreoretinal interface and their dynamics over time by means of changes in number, size, intensity, eccentricity and orientation.

## 2. Materials and Methods

### 2.1. Data Collection

Images of the vitreoretinal interface were obtained, using OCT (HR-OCT Spectralis, Heidelberg^®^ Engineering, Germany), in previous interventional studies carried out by our research group detailing the methodology used for the generation of two different animal models of chronic glaucoma (MEPI and Ms) [50]. The experiment was previously approved by the Ethics Committee for Animal Research (PI34/17) of the University of Zaragoza (Spain) and was carried out in strict accordance with the Association for Research in Vision and Ophthalmology’s Statement on the Use of Animals. The widely used Morrison model [11] was generated with sclerosing injections of episcleral veins (MEPI) of the right eye of Long–Evans rats, performed biweekly for 6 months. The second model (Ms) was generated by injecting a 2 microlitre suspension (10% w) of 20/10 biodegradable PLGA [51] microspheres into the anterior chamber of the right rat eye at 0, 2, 4, 8, 12, 16 and 20 weeks. The left eyes did not undergo intervention. IOP (with Tonolab^®^ rebound tonometer) and OCT scans of both eyes were performed at 0, 2, 4, 6, 8, 12, 18 and 24 weeks. A cohort that did not undergo intervention served as control and was scanned at 0, 12 and 24 weeks.

### 2.2. Image Analysis

Images were acquired using a high-resolution OCT device (OCT Spectralis for research on animals) with a plane power polymethylmethacrylate contact lens (thickness 270 μm, diameter 5.2 mm) (Cantor+Nissel^®^, Northamptonshire, UK) adapted to the rats’ cornea to obtain higher quality images [52], in every scan examination. This neutral dioptric power lens (no positive or negative dioptric addition) allowed continuous hydration of the cornea preventing desiccation and kept the refractive state stable during the examination. The retinal posterior pole protocol with automatic segmentation, eye-tracking software and follow-up application were used to ensure that the same points were re-scanned throughout the study. The “Enhance depth imaging” mode was disabled in all cases.

The OCT images were exported as Audio Video Interleave (AVI) videos. These videos were composed of cross-sectional images acquired by means of 61 b-scans measuring around 3 mm in length and centred on the optic nerve. These cross-sectional images are 3 mm in length and 0.969 mm in height, so the image area is 2.906 mm^2^. Image resolution was 3 microns per pixel generated. Each cross-sectional image has a total of 1536 × 496 pixels. Therefore, each pixel has an area of 3.815 µm^2^. These videos were analysed using a custom program implemented in Matlab (version R218a, Mathworks Inc., Natick, MA, USA). The imaging data were analysed by a masked reader. OCT segmentation was performed by two different researchers, likewise masked, to verify reproducibility.

Vitreous/retinal pigment epithelium (VIT/RPE) relative intensity was quantified as an indirect measure of immune response [44,48,49]. Our custom program finds the internal limiting membrane (ILM) and the inner and outer layers of the RPE by greyscale conversion, obtaining the segmentation of vitreous and RPE (Figure 1). The vitreous and RPE intensity value was calculated as the mean of the intensity of all pixels within each region, giving the VIT/RPE relative intensity in each b-scan. Thus, the VIT/RPE relative intensity of each eye is the mean of the 61 b-scans.

The vitreous opacities, closely related to the immune cells, were analysed. This technique focuses on the analysis of hyperreflective opacities in the vitreoretinal interface using OCT, which does not require a correction factor for histological correlation [47] and ensures characterization of the actual opacity. These opacities were studied according to their size, taking as a reference previous studies of morphological analysis of microglia in the retina and histological analysis of hyalocytes [53]. It is possible to discriminate between non-activated and activated cellularity according to the size of the soma, since microglia morphology differs depending on their state of activation. The smallest cells (early growth) have a rounded or amoeboid morphology. Resting cells (non-activated) are characterized by a thin cell body with ramified cellular processes; and reactive cells (activated) have a larger somatic size with phagocytic activity and motility [28,31].

Individual hyperreflective opacities, extrapolated as soma areas, were automatically measured and classified into groups according to size: isolated cells (<10 μm^2^), non-activated cells (10–50 μm^2^), activated cells (50–250 μm^2^) and cell complexes (>250 μm^2^). Our custom program determined the size of the opacities in each b-scan by calculating the number of pixels in each opacity. It is necessary to convert the greyscale image to binary image in order to identify all regions in the vitreous. First, the greyscale image was filtered with a multidimensional filter that emphasizes horizontal edges by approximating a vertical gradient; and second, the filtered image was converted to a binary image using the global image threshold, specified as a scalar luminance value, determined by Otsu’s method [54]. This method chooses a threshold that minimizes the intraclass variance of the thresholded black and white pixels. To ensure the measurement, the background speckle noise was deleted using a denoising filter. Using grey scale and binary images, this filter distinguishes between hyperreflective opacities and background noise because the opacity intensity is greater than the background intensity (see Figure 1). In our custom denoising filter, individual hyperreflective opacities correspond to areas whose intensity exceeds the noise intensity. To quantify that, we computed the upper outliers of the intensity of the areas in the vitreous for each cross-sectional image. Those areas with intensity lower than this limit were removed, while the most reflective areas were extrapolated as soma areas. Thus, only the cellularity was quantified, eliminating the physiological phenomena of the eye [55]. In this way, opacity intensity was obtained from the grey scale image using the binary image as a mask to detect the regions. Moreover, number, size, eccentricity and orientation of these opacities were obtained from the binary image.

Once each opacity was detected, several parameters could be computed for each eye at different stages of follow-up. As a measurement of the overall immune response to the induced model vs. physiological conditions, the total cell area was calculated by the number of opacities and the area of each opacity. The mean number of opacities was an indicator of immunity to noxa over time that allowed analysis of in situ resident immune cellularity and intra- or extra-ocular recruitment [56,57,58,59]. The mean area of opacities was calculated for all cells and for each group according to cell area, obtaining reliable reproducibility of the cell soma. By computing the cell percentage for each group, we studied the dynamic relationship between activated and non-activated cell populations.

The mean intensity of opacities is of great importance since immune activation involves gene-protein expression prior to soma remodelling. This parameter is the mean of the intensity values of each pixel in the opacity. Eccentricity is the ratio of the distance between the foci of an ellipse and the length of its major axis. An ellipse with an eccentricity of 0 is actually a circle, while an ellipse with an eccentricity of 1 is a line segment. Therefore, the eccentricity indicates if the cell morphology is more linear, elongated or flat (values closer to 1) or, on the contrary, more rounded (values further from 1). Finally, the orientation of opacities was analysed as a quantification of the motility or active displacement of immunity towards the noxa [28,33,35,60,61]. This orientation is the angle between the *x*-axis and the major axis of the corresponding ellipse of each opacity and its value is represented in degrees.

### 2.3. Histological Analysis

To investigate the nature of vitreous hyperreflective opacities observed by OCT during glaucoma, six rat eyes, in three of which episcleral veins were sclerosed producing ocular hypertension (>20 mmHg) for 24 weeks, were analysed. Paraffin-embedded eyes were sectioned (3 µm) along the eye axis, deparaffinized and rehydrated. After several washes in phosphate buffered saline (PBS), sections were incubated overnight at 4 °C with goat anti-human Iba1 (Abcam, Cambridge, UK) at 1:100 dilution. After washing the sections in PBS, they were incubated for 2 h at room temperature with rabbit anti-goat Alexa 568 (Invitrogen, Carlsbad, CA, USA). SYTOX Green Nucleic Acid Stain (Invitrogen) diluted in PBS (1:500 dilution) was incubated for 10 min for nuclear counterstaining. Slides were mounted in Fluoromount (Sigma-Aldrich, St. Louis, MO, USA) medium for further analysis using a laser scanning confocal microscope (TCS SP5; Leica Microsystems GmbH, Heidelberg, Germany). Immunohistochemistry controls were conducted by omission of the primary antibody in a sequential tissue section. Hematoxylin/eosin stain was also performed to study the morphology and presence of intravitreal cells on the surface of the retina.

### 2.4. Statistical Analysis

All data were recorded in an Excel database, and statistical analysis was performed using SPSS software version 20.0 (SPSS Inc., Chicago, IL, USA). The variables under study were as follows: eye (intervened right eye vs. non-intervened left eye), sex (male vs. female), type of chronic glaucoma model (MEPI vs. Ms) and control, number of injections performed per model, intraocular pressure and vitreous signal intensity (VIT/RPE) measured using OCT.

After checking for variable normality with the Kolmogorov–Smirnov test, we performed parametric test using multiple comparisons by ANOVA and Tukey post-hoc analysis (to identify between which groups there are statistical differences) and analysis of correlations with Pearson’s *p* test. All values were expressed as means ± standard deviations. Although values of *p* < 0.05 were considered to indicate statistical significance, but also the Bonferroni correction for multiple comparisons was calculated to avoid a high false-positive rate.

## 3. Results

### 3.1. Descriptive Data

A total of 271 OCT videos, extracted from 95 animals (40% males/60% females) at different times of study follow-up, were analysed. Episcleral model (*n* = 35 animals): 72 videos from the right eye (RE)/47 videos from the left eye (LE); Ms model (*n* = 28): 38 RE/26 LE; healthy controls (*n* = 32): 31 RE/57 LE. The number of eye injections inducing each glaucoma model, and the IOP curves they generated compared to healthy controls, are shown in Figure 2. Glaucomatous and healthy males had higher IOP levels than females throughout the study (data extracted from [50,62]).

### 3.2. Descriptive Data

OCT analysis of the vitreous detected higher vitreous/retinal pigment epithelium (VIT/RPE) intensities in chronic glaucoma. After the first hypertensive injection, the MEPI model presented the highest initial vitreous signal intensity value, coinciding with the greatest initial fluctuation in IOP increase. This trend was maintained until 12 weeks (Figure 3a). The Ms model presented lower initial vitreous signal intensity, equalled the MEPI model at week 8 (even when IOP still remained at ocular normotension levels (<20 mmHg)) and, from week 12 onwards, the Ms model surpassed the MEPI model. Non-injected left eyes also showed a slight increase in vitreous signal with respect to healthy controls (Figure 3b). Healthy control animals’ IOP and vitreous signal intensity curves showed lower levels than both chronic glaucoma models (Figure 2 and Figure 3).

In addition, the influence of sex on vitreous signal was analysed. In general, females of both chronic glaucoma models showed slightly higher VIT/RPE OCT intensity than males and their healthy female counterparts. However, under physiological conditions (healthy control) males showed a peak of vitreous intensity at week 12 (16 weeks of life) that declined in later phases of the study (Figure 3c,d).

### 3.3. Correlation Analysis

A correlation study was performed to determine the influence of the model on the VIT/RPE intensity analysed using OCT, as a marker of immunity. The MEPI model generates higher early IOPs (but without intraocular injection) than the Ms model, which has slower and progressive IOPs but is induced by intraocular injections with rupture of the ocular barrier and, therefore, induction of anterior chamber associated immune deviation (ACAID) [19,63]. The most relevant results and the strongest statistically significant correlations are shown (in bold) for all animals (Table 1) and by sex (Figure 4).

#### 3.3.1. MEPI Model

Males presented higher IOPs at 4 weeks (r = 0.713, *p* = 0.014) and 6 weeks (r = 0.759, *p* = 0.007) as the number of injections increased (Figure 2). This correlation was not found in females. Males with higher IOPs after the first injection (2 w) also showed a later correlation (6 w) with higher IOPs (r = 0.832, *p* = 0.001) (Figure 4). In both sexes, week 6 OCT intensity correlates with week 12 intensity (r = 0.762, *p* = 0.001) (Figure 4). In the episcleral model, the presence of inflammation detectable by vitreous OCT signal at week 6 was maintained and even increased at later stages.

#### 3.3.2. Ms Model

In this model, both sexes presented a moderate direct correlation in IOPs between early (Figure 4d) and intermediate stages of the study (r = 0.716, *p* < 0.001) (Figure 4e). This became strong between weeks 8 and 24 (milestone reached later than in MEPI 2 w/24 w) (Table 1). However, in this model, an inverse correlation was found at week 2 between IOP and OCT (r = −0.775, *p* = 0.024) (Figure 4f), between IOP 2 w and OCT 12 w (r = −0.999, r = 0.026) and between IOP 12 w and OCT 18 w (r = −0.998, *p* = 0.045), mainly at the expense of females. Even when IOP did not reach ocular hypertension values (lower IOP levels) (higher) vitreous OCT signal was generated in that same week and later.

#### 3.3.3. Healthy Control Cohort

Physiologically, a positive IOP correlation at early stages of the study (IOP4w-IOP8w r = 0.934, *p* = 0.020) was observed in both sexes (Table 1 and Figure 4g). Furthermore, in females, an inverse correlation IOP baseline -IOP 6 w (r = −0.999, *p* = 0.021) and direct correlations between IOP and OCT at intermediate (IOP 12 w-OCT 12 w; r = 0.997, *p* = 0.049, in the left eye) and late stages (IOP 18 w-OCT 24 w; r = 0.854, *p* = 0.031, in the right eye) were observed (Figure 4h). In females, the age-related degenerative process (possible neoepitope generation) [64,65] produces higher vitreous OCT intensity (reflex of immune involvement and/or activation) correlated with IOP even in a situation of ocular normotension. However, in males, moderate inverse correlation was found at the end of the study (IOP 24 w-OCT 24 w; r = −0.766, *p* = 0.045). In healthy control animals, no statistically significant strong correlation was found between OCT intensities, suggesting lower inflammatory influence under physiological conditions compared to glaucoma conditions.

Analysis of VIT/RPE signal intensity (as an indirect measure of vitreous parainflammation) showed different behaviour according to glaucoma model, inducing injections and sex. IOP levels correlated with vitreous intensity using OCT directly at early and late stages of the study. In addition, contralateral non-induced eyes also showed higher vitreous signal using OCT than healthy controls.

### 3.4. Histology

Based on the hypothesis of considering vitreous OCT intensity a reflection of the activity of the vitreous immune population, histological studies were performed to cellularly characterize the hyperreflective opacities.

An increased number of cells in the vitreous cavity was observed in the glaucomatous eyes (Figure 5). This was in accordance with a previous paper showing that the number of intravitreal cells was higher when IOP was experimentally elevated in adult mice [34]. Preretinal localization of intravitreal cells close to the internal limiting membrane, as well as their round shape, suggested that these cells could be hyalocytes [66] (Figure 5). Hyalocytes exhibit many phenotypic characteristics of macrophages and have a well-described phagocytic function that likely aids in maintaining vitreal transparency through the clearance of cellular and extracellular debris [67]. Hyalocyte-like cells in hypertensive rat eyes showed positive for ionized calcium-binding adaptor molecule 1 (Iba1) (Figure 6), a 17-kDa protein whose expression is restricted to macrophages [68], including hyalocytes [41]. As expected, the Iba1 antibody also marked activated microglia embedded in the retina of hypertensive eyes (Figure 6) [69]. Similarly, as occurs in other models of ocular inflammation [41], examination of the ciliary body in hypertensive eyes revealed that the ciliary body was surrounded by many hyalocyte-like cells (Figure 6), some of which seemed to have migrated from the ciliary body, crossing its outer non-pigmented epithelium (Figure 6). This could suggest that, as has been postulated by other authors [41], the ciliary body could be a potent source of vitreal macrophages during eye inflammation. Hyalocyte-like cells in hypertensive rat eyes were never related with protruding glial membranes that, from the internal limiting membrane, penetrate the vitreous body (Figure 5 and Figure 6), suggesting that the migration of microglia through the internal limiting membrane as proposed by Santos et al. [70] in a model of photoreceptor degeneration would not be a common phenomenon in hypertensive rat eyes. However, this cannot be ruled out.

Once the existence of vitreous changes detected by OCT (VIT/RPE signal) was evidenced in glaucoma models, and the vitreous hyperreflective opacities were corroborated by histological studies as cell-like Iba-1+ hyalocytes with phagocytic activity, the vitreous signal was studied as an indirect representation of the immune population by analysing the size of the hyperreflective opacities [31,33].

### 3.5. In Vivo Analysis of Vitreous Immunity Detected Using OCT

The individual hyperreflective opacities detected using OCT and characterized as Iba1+ cell-like hyalocytes were analysed. As a representation of overall immune response, the total area of opacities/cells was quantified. In the glaucoma models, right eyes showed significantly increased total areas (MEPI > Ms), and left non-induced eyes showed increase (no significantly) compared to healthy controls (Figure 7). To ascertain whether the increase in total cell area was due to an increased number of cells and/or an increase in cell size, and thus an increase in activated cells, the following was performed:

#### 3.5.1. Quantification of the Mean Number of Opacities/Cells

Under physiological conditions, the number of opacities remained at 10–20; however, glaucoma-induced right eyes had a higher number of hyperreflective opacities/cells that fluctuated with a mean of approximately 70 opacities/cell. The highest quantification occurred early in the MEPI model. The Ms model showed an initial peak, coinciding with the first intraocular injection, and achieved elevated levels at later stages. The left eyes from both models also showed a slight increase in cell number (Figure 7). The results of these two in-depth analyses are in agreement with previous results for relative VIT/EPR intensity [49].

#### 3.5.2. Percentage of Opacities/Cells by Size 

The set of hyperreflective opacities or vitreous cell populations was divided into isolated cells (<10 microns^2^), non-activated cells (10–50 microns^2^), activated cells (50–250 microns^2^) and cell complexes (>250 microns^2^) [31,33] (Figure 8) with the aim of determining the changes in the population ratio in the non-activated and activated state in glaucoma vs. healthy eyes (Figure 9). Study at the vitreoretinal interface does not require a correction factor and the histological similarity can be measured directly [47]. Under physiological conditions, a population ratio ordered from lowest to highest representation was found: isolated cells (<10 microns^2^) < complexes (>250 microns^2^) < activated cells (50–250 microns^2^) < non-activated cells (10–50 microns^2^). Under glaucoma conditions, a specular response was found between opacities of 50–250 microns^2^ (activated cells) and 10–50 microns^2^ (non-activated cells), respectively. Dynamic fluctuations were observed, but approximately 40–50% on average were opacities >50 microns^2^ in size (activated cells). In addition, the onset of damage generated an early peak in complexes (>250 microns^2^). The Ms model (induced by intraocular injection) produced a higher proportion of opacities in the 50–250 microns^2^ range, suggesting early vitreous activation.

#### 3.5.3. Mean Intensity of Opacities/Cells

Under physiological conditions, the lowest intensity was quantified in isolated opacities/cells (<10 microns^2^) and progressively increased with size: opacities of 10–50 microns^2^ (non-activated cells) followed by opacities of 50–250 microns^2^ (activated cells). However, under glaucomatous noxa, this trend reversed and a greater variation in intensity was quantified in the smallest opacities/cells (<10 microns^2^) (Figure 10) prior to the increase in size. Subsequently, with increased soma size (activated cells with pseudopod formation) [40,53,67,71], a relative decreased mean intensity was quantified.

#### 3.5.4. Average Eccentricity of the Opacities/Cells

This analysis allows observation of cell morphology. Linear or flat morphology (eccentricity close to 1) vs. rounded morphology (eccentricity far from 1). Under physiological conditions, isolated opacities/cells (<10 microns^2^) presented the most rounded or amoeboid morphology (eccentricity 0.85) as opposed to opacities/cells with progressively larger sizes of 10–50 microns^2^ (non-activated), followed by 50–250 microns^2^ (activated cells) and <250 microns^2^ (cell complexes), these being increasingly flat (eccentricity 0.95–1). Under conditions of glaucomatous noxa, this same trend was maintained. However, at early stages of the study, mean eccentricity was even lower than under healthy conditions (and was lower in the Ms model with intraocular injection) (Figure 11).

#### 3.5.5. Mean Orientation of the Opacities/Cells

Orientation was analysed to discover if examination of the vitreous OCT made it possible to objectify an active shift (change in mean orientation) of immunity towards the noxa [35,41,60,72]. Under physiological conditions, stability was observed in all sizes of opacities/cell groups. However, under glaucomatous noxa conditions, the smaller opacities (<10 microns^2^: isolated ovoid cells) of both models showed an early change in orientation. In addition, the Ms model showed a progressively increasing change in orientation that continued until the later stages of the study (Figure 12). Continuous activation and motility of early vitreous immunity was detected prior to the increase in opacity/soma size, which coincided with the increase in IOP and intraocular injections.

## 4. Discussion

Imaging study: OCT is a fast, non-invasive tool that provides repeatable analysis of the neuroretinal structure and measurement of different retinal layers [73,74]. However, it has the handicap of not being able to identify the different cell types within the neuroretinal thickness, for example, to differentiate between immune cells and the rest of the neuronal, supporting or vascular cells. Studies that have identified certain cell types either require animals genetically modified to express fluorescence or are technically very complex [75]. OCT is a technology capable of offering almost histological images depending on the light transmitted or reflected as it passes through different structures with differing densities. It can be utilized to identify tissue structure or composition. With the development of more advanced technologies, we have made the leap to vitreous analysis under both normal and inflammatory conditions. The size of the vitreous cells involved (10 microns) is larger than the resolution of OCT (3 microns), which allows the number of cells to be quantified manually or, in our case, by automated image-counting methods [76]. In addition, OCT has the added advantage of having its peak sensitivity in the vitreous over the retina. Evaluating opacities on a background with less diffuse backscattering due to tissue homogeneity of the vitreous (mainly water) allows us to analyse the intensity changes more reliably than in the retina (with tissue heterogeneity and hundreds of neurons, connecting cells, astro/microglial and blood cells). Those areas that light finds it harder to pass through have higher optical densities (intensity in our study) corresponding to lipid or calcium membranes. Related to this idea, Keane et al. [77] published a study demonstrating that the relative intensity of VIT/RPE correlates positively with clinical vitreous turbidity. This technique was validated in subsequent studies [46,78]. Rodrigo et al. [49] described greater vitreous signal in rats with glaucoma than in healthy rats. Recently, a study performed in humans using OCT identified ramified macrophage-like cells on the retinal surface, suggesting a hyalocyte origin (but without histological confirmation), with altered morphology in patients with retinopathies and associated with areas of retinal nerve fibre layer (RNFL) thinning in glaucoma patients. However, this technique is limited by complexity, long acquisition time, axial length correction to correct for ocular magnification, manual cell identification and high cost [72]. To our knowledge, vitreous OCT analysis has not been used in animal research to analyse in depth the vitreous parainflammation caused by different models of ocular hypertension.

Parainflammation is an intermediate state between the basal homeostatic state and the classic inflammatory response, which is more common but of lesser magnitude. An immune response is initially beneficial and necessary to restore tissue homeostasis and promote tissue cleaning, healing and functionality. Glaucoma results in neuroretinal damage and, when mediated by glutamate, damage-associated molecular patterns released by RGCs and glia trigger parainflammatory responses [28]. Retinal response to glaucoma damage appears to be neuroprotective and mediated by myeloid-derived suppressor cells [56]. In the event of damage, astrocytes and retinal microglia release pro- and anti-inflammatory factors [79] capable of activating and regenerating retinal progenitor cell niches [56]. Both are activated at the same time and negotiate the removal of waste, this falling primarily on microglia. However, when microglia are not sufficient, astroglia compensate, albeit in a delayed manner leading to pathological conditions [60]. If there is a defect in immune response pathways due to accumulating risk factors, prolonged or sustained inflammatory stimulation, or to “neo-antigens” generated with ageing, the physiological equilibrium may be impaired and the regulatory mechanisms are altered [75], thereby converting beneficial immunity into a neurodestructive autoimmune process [80]. Microglia and macrophages undergo important dynamic morphological changes in their function. In an inactive situation, they are branched to serve as a sensor to changes in the microenvironment. Upon damage they are activated, upregulate surface and complement receptors, secrete soluble factors and undergo changes in shape, proliferation, migration and phagocytosis through rearrangements of the actin cytoskeleton. The Iba1 protein [69] is a calcium- and actin-binding protein for cell migration and phagocytosis of hematopoietic cells, restricted to microglia and macrophages; therefore, its staining/marking rules out other cell types such as lymphocytes, astrocytes or oligodendrocytes. Microglial activation is one of the first events in glaucomatous neuronal damage that occurs prior to RGC death; low levels of microglial activation induce delayed neurodegeneration and control of its activation reduces optic nerve damage [28].

Vitreous and hyalocytes: The inhibitory vitreous microenvironment is regulated by hyalocyte-like macrophages and mediated by soluble molecules [81]. The vitreous medium, with high water content embedded in the extracellular matrix, is ideal for the transmission of soluble factors, meaning that hyalocytes can detect changes in the microenvironment [82] easily and rapidly to target the noxa. Hyalocytes present secretory granules, lysosomes, a developed Golgi apparatus, a lobulated nucleus, cytoplasmic projections and a moderate number of mitochondria [40]. In a state of activation secondary to changes in the microenvironment, the number/size of intracellular organelles increases and, therefore, the membrane content increases, facilitating its possible detection by imaging tests. Under normal conditions, the vitreous is transparent. Analysing a structure designed to be virtually invisible (vitreous) by non-histological imaging techniques is difficult. Dark-field slit microscopy revealed many spots in the cortex of vitreous that scatter light intensely, the small ones being described as hyalocytes and the larger ones as debris [38]. In the activated state, increased intracellular membranes and organelles may generate increased hyperreflectivity with OCT. A two-week study using OCT detected a higher density of vitreous opacities located near areas of cell death [48]. The nature of these vitreous opacities was unknown and it was posited they were inflammatory cells, cellular debris or other factors migrating through the vitreous. In our 24-week follow-up study, these opacities were analysed in depth and a hyalocyte origin was demonstrated.

Characterization of hyalocytes under physiological and pathological conditions remains unclear.

Under physiological conditions, there is a slow turnover with long half-lives of macrophages resident in the brain, retina [83] and vitreous [40,67]. In a histologic study of healthy guinea pig eyes, Ogawa K [53] describes free hyalocytes of elongated morphology (approximately 50 microns) entangled in the vitreous cortex, and the highest density of small, mostly ovoid, free cells (approximately 20 microns) on the surface of the ciliary body. The abundance in the ciliary body epithelium suggests that it is a site of precursor emigration from the ciliary stroma, consistent with our results (Figure 6). Hyalocytes can enter the vitreous through the ciliary body epithelium and subsequently move along the longitudinal tracts of the vitreous gel into the neuroretinal vicinity. In mice, hyalocytes are concentrated near the papilla and at the vitreous base, with the latter location coinciding with a higher concentration of total proteins [84]. In rats, 90% presented macrophage markers and 15% monocyte/macrophages, which is close to the percentage found in healthy rats in our study, where approximately 95% were in the large cell range (10–250 microns) and approximately 5% in the smaller cell range (<10 microns—monocytes). The higher proportion of vitreous opacities/cells in healthy rats coincided with opacities of sizes corresponding to non-activated (or anti-inflammatory state) cells, consistent with the eye’s immune inhibitory privilege state [81,85] and with the control microglia size (20–60 microns^2^) identified in retinal studies [33]. A variable number of opacities corresponding to cells activated for the maintenance of homeostasis was also quantified [35] (Figure 9).

Sex: The healthy females in our study showed a linear vitreous signal, unlike the males, in which at around 16 weeks of age (12 weeks of the study) vitreous activation seems to occur, coinciding with an increase in retinal thickness and increased remodelling with subsequent loss [62]. In this regard, variations in microglial density between males/females have been described in different brain areas such as the preoptic area [61], and marked sexual dimorphism has been found in mouse peritoneal macrophages [86]. Hyalocytes share phenotypic characteristics with peritoneal and ependymal macrophages as serous cavity roving macrophages. The replacement rate of peritoneal macrophages was very high in males at 16 weeks of age, after sexual maturation, but was decreased and slow in females, coinciding with our results (Figure 3c,d). In contrast, immune disturbances tend to be more frequent in females, but more severe in males [87,88,89]. Females have demonstrated an enhanced ability to control infection [86] and less neuroretinal tissue loss in the presence of glaucomatous noxa [90]. The greater vitreous intensity observed in females in the glaucoma models, as well as in healthy animals at later ages (28 weeks of age), suggests a greater immunogenic presence that could act in an initially reparative manner in the presence of hypertensive noxa or age-associated neoepitopes, maintaining retinal structure and function [62,90]. These results highlight the importance of considering the influence of sex (traditionally not taken into account) and age in immune research, due to susceptibility to infectious, inflammatory or age-associated diseases such as glaucoma.

Visualization of neuroinflammation in the CNS and eye: In early stages of chronic neurodegenerative and psychiatric diseases, neuroinflammatory mechanisms initiated by chronic activation of innate immunity alter function and trigger death. As microglial alterations (dynamic changes in density and activation) are detectable in early stages of disease and precede neurodegeneration, they have been established as diagnostic and therapeutic biomarkers of progression [61,91]. Direct visualization of the recruitment of specific immune cells from the periphery to the brain [57] currently requires expensive and invasive techniques such as 2-photon microscopy (T cells), MRI (macrophages) and PET and SPECT (microglia). In the retina [33], in vivo microglia tracking using CSLO imaging has been achieved [92], allowing them to be evaluated independently of other retinal cells. This technique requires animals genetically modified to express fluorescence, and for obvious reasons cannot be performed on humans; however, analysis of the vitreoretinal interface using OCT is now within the reach of conventional ophthalmologic clinics. In addition, the image quality of the Bosco et al. technique [33] is critically dependent on background autofluorescence. In contrast, our vitreous OCT measurement filters and removes smaller-sized cellular elements and therefore ensures correct counting without interference. Comparing the retinal findings from the CSLO microglia study with the vitreous findings from the OCT hyalocyte study, we found comparable cell percentages, although with some key differences. Using CSLO analysis of retina in mice with glaucoma, 200–300 total microglial cells were quantified, and of these 10–180 were activated (approximate mean of 27–34%, with large variation of 5–60%). In our study, a mean of 70 opacities/cell (approximately 40–100) were counted, with a higher activation percentage (approximate mean of 40%) and lower variability (30–50%). It was not possible to analyse histologically whether hyalocyte activation was precocious to retinal microglia, but the higher percentage of vitreous opacities (hyalocytes) in the cell activation range found in our results with respect to retinal cells (microglia) suggests it was. It could thus serve as an indicator of progression or be linked to initial events in glaucoma pathogenesis.

Analysis of the vitreous in two glaucoma models.

The episcleral model alters the aqueous humour outflow pathway by sclerosis of episcleral veins via retrograde hypersaline injection; consequently, the blood–ocular barrier would be altered and activate the perivascular glia [16,93], triggering the inflammatory response. The MEPI results suggest that those animals with the lowest initial IOPs will have the highest IOPs after the onset of repeated episcleral hypertensive stimuli. In this model, the main correlations occur at early stages. The first hypertensive injection triggers an increase in vitreous intensity. The more damage produced at the first injection, the higher the IOP at week 2, which correlates with the OCT-detectable vitreous signal (r = 0.812, *p* = 0.027). Greater and earlier involvement of retinal glia was observed by histology in previous studies [50,51]. In addition, in left eyes an inverse correlation was found between baseline IOP levels and vitreous OCT 18 w (r = −0.979, *p* = 0.004), reflecting possible higher late contralateral inflammatory vitreous activation in those more hypotensive (susceptible) non-induced eyes, and also higher vitreous OCT signals the higher the IOP at later stages, IOP 24 w-OCT 24 w (r = 0.885, *p* = 0.008).

The Ms model triggers an ACAID response [85] to trauma (intraocular injection in the anterior chamber). After cannulation and acute IOP increase, a marked microglial response was observed, with similar increases in vitreous hyalocytes [34], coinciding with our observations of increased vitreous signal with cannulations (Figure 7). The Ms model presented a strong positive correlation in IOP at intermediate stages (IOP 8-24 w: r = 0.917, *p* = 0.028), later, therefore, than the episcleral model, where this milestone occurred from 2 w onwards (IOP 2 w-24 w). Likewise, correlation was detected in the non-induced left eye at IOP 4 w-12 w (r = 0.800, *p* = 0.004) and IOP 18 w-OCT 18 w (r = 0.890, *p* = 0.043), also demonstrating contralateral vitreous inflammatory activation.

Contralateral involvement: In the retina of a glaucoma mouse model with ocular hypertension induced by cauterization, microglial activation was observed in both the induced eye and the contralateral eye, exhibiting lower intensity, but which remained stable over time [94]. Our study produces similar results in the analysis of vitreous opacities/hyalocytes obtained in models of more progressive and chronic hypertension. The contralateral eye also shows vitreous changes with respect to the healthy control. These appear to be resident hyalocytes (without recruitment), as an increase in intensity (Figure 3b) but hardly any increase in number (Figure 7) was detected. Other glaucoma studies examining retinal microglia have shown contralateral activation even at normal IOP values, suggesting that increased IOP may not be a contributing factor in early stages [29], and with the superior colliculus acting as a communicative structure [12]. Individual susceptibility to increased IOP may differ based on the ability to control the immune response, and each individual’s immune background may modify disease progression. Our vitreous OCT study is also able to detect early changes even in conditions of ocular normotension.

Affectation prior to IOP increase (normotensive values): Changes in gene expression related to immune response [95], acute stress and proinflammatory markers [12] have been demonstrated with early cytokine dysregulation, independently and prior to IOP increase, detection of RGC loss and axonal degeneration [28,79]. Cooper et al. [95] demonstrated activation of the complement cascade (plasma proteins to opsonize cellular debris by formation of membrane attack complexes, as a bridge between the innate and adaptive response) in retinal degeneration with increased inflammatory response that facilitates apoptotic cell elimination and Iba1 activation at an early stage of IOP increase. It was suggested that IOP increase could later act as an aggravating factor that exacerbates disease progression. In addition, Tsai et al. [14] described, in a model of experimental autoimmune glaucoma, an increase in complement activation after 6 weeks of IOP induction. Our results also showed an increase in larger vitreous opacities (>250 microns^2^) at 6 weeks (Figure 9) that could correspond to complement-opsonized apoptotic bodies.

Analysis of vitreous opacities as a biomarker in glaucoma: There is still a long way to go in understanding the actions of immune cells in both healthy and diseased eyes. In the retina, microglia are considered activated when the soma is >50 microns. However, our results detect increased intensity (onset of intracellular machinery) in the vitreous and a change of orientation (onset of displacement) as an expression of activation in the smallest opacities at an early stage. Our findings suggest the smallest cells (<10 microns^2^) would be the first in the communication and transformation chain to detect a change in homeostasis or noxa detection in the vitreous. This suggests that small cell analysis could serve as an early marker of vitreous immune activation, prior to the morphological change of the soma. In the episcleral model, the change is mainly observed in the initial hypertension. However, in the Ms model, even without an IOP increase, a marked change of orientation is generated with a progressive increase indicating that more and more cells are oriented towards a certain point (retinal damage). However, at later stages, the mean orientation returns to 0, suggesting that there could be more neuroretinal areas with damage in different locations and, therefore, more orientations to be taken by the cells, annulling the summation effect of the mean orientation. On the other hand, prior studies have also described how in the brain and retina the motility capacity of microglia damaged by chronic stress decreases [61]. In the somatosensory cortex, dystrophic microglia was compensated by microgliosis (increased number of microglia). Accordingly, another possible explanation of our results from both models, concerning the return to 0 orientation of the opacities/hyalocytes, could indicate “ageing or motility damage” due to accumulation, repetition and chronicity of oxidative stress. Reinforcing this idea, in the MEPI model, an increase in the number of opacities/hyalocytes is observed (Figure 7) from week 8 onwards (when the mean orientation is cancelled) (Figure 12). However, in the Ms model, the change in orientation is maintained along with the variation in the number of opacities/hyalocytes, possibly being secondary to the multiple intraocular injections. It has been shown that every intraocular injection (at least in the anterior chamber of the rat eye) triggers vitreous parainflammation, and repeated injections exacerbate this immune response with subsequent neuroretinal damage [33,50,92].

In the event of disease, reactive microgliosis is induced, which may be dependent on infiltration [96] and/or proliferation by self-renewal of adult CNS resident cells [58]. Both systemic and local exposure to harmful molecules can initiate the influx of myeloid cells into the vitreous. Vagaja N et al. [41] demonstrated in mice that after systemic noxa with lipopolysaccharide to simulate infection, an accumulation of hyalocytes is produced in the posterior margin of the ciliary body and inner retinal surface; and after multiple injections, there is much greater abundance in apical ciliary processes. London et al. [56] demonstrated that after glutamate intoxication insult, there is recruitment and activation of innate immune cells in retina on days 1 and 2; however, the increase in IOP produced a milder increase in immune cells. Before macrophages are observed in the retina, there is a proinflammatory profile, which changes to anti-inflammatory. Monocyte-macrophages are essential as anti-inflammatory and neurotrophic elements to support cell renewal of progenitor cells in the ciliary body and contribute to the initiation or resolution of inflammatory and regenerative states by switching functions [28]. Our vitreous opacities/hyalocytes results also show this specular relationship and anti/proinflammatory dynamism (Figure 9). In cell renewal, the primitive or more immature cells are rounded and, as they mature and acquire capabilities, they change their phenotype and morphology to less rounded [40,97]. The eccentricity results show that the Ms model, which has a higher proportion of more rounded or early cells, triggers a call for a higher number of primitive cells (Figure 11) from the ciliary body (Figure 6), or from extraocular circulation. However, the higher number of opacities/hyalocytes quantified in MEPI (Figure 7) suggests that they belong to those located in situ intraocularly.

Limitations and future perspectives: Circulating immune cells from the periphery can rapidly enter the CNS under pathological states and conditions that alter the blood–brain and blood–ocular barrier [79]. The limitations in the histologic phenotyping of our study could not resolve the dilemma of the infiltrative origin [98] (lymphocytic or not) of the increased vitreous opacities.

The authors are aware that many aspects and unknowns remain to be resolved. In our opinion, however, this finding paves the way for in vivo analysis of the immune side of glaucoma (of great importance in normotensive glaucoma and in the progression of glaucoma, even with normal pressure levels) in a non-invasive (without animal sacrifice, histological studies or contrasts), simpler and possibly earlier way than examining astroglial changes [99] in the retina, which are very difficult to detect nowadays.

There is a need to monitor pathological processes and evaluate the therapeutic efficacy of anti-inflammatory and other potential neuroprotective drugs [100]. Designing therapies for glaucoma other than hypotensive mechanisms would be beneficial and neuro-immunomodulatory targeting is emerging as a key to effective new therapies. Mesodermal immune cells could be important candidates for therapeutic interventions without direct impact on neuroectodermal lineage cell types [61]. Our method of analysing vitreous opacities/hyalocytes using OCT could serve as a reliable neuroimaging biomarker to detect disease onset and early progression applicable to glaucoma and potentially other neurodegenerative diseases impacting the retina and optic nerve [101].

## 5. Conclusions

Glaucoma presents chronic and subclinical inflammation. In a novel way, this study detects and monitors parainflammation non-invasively using optical coherence tomography of the vitreous, while computational analysis characterizes immune cellularity based on morphology.

## Figures and Tables

**Figure 1 biomedicines-09-01792-f001:**
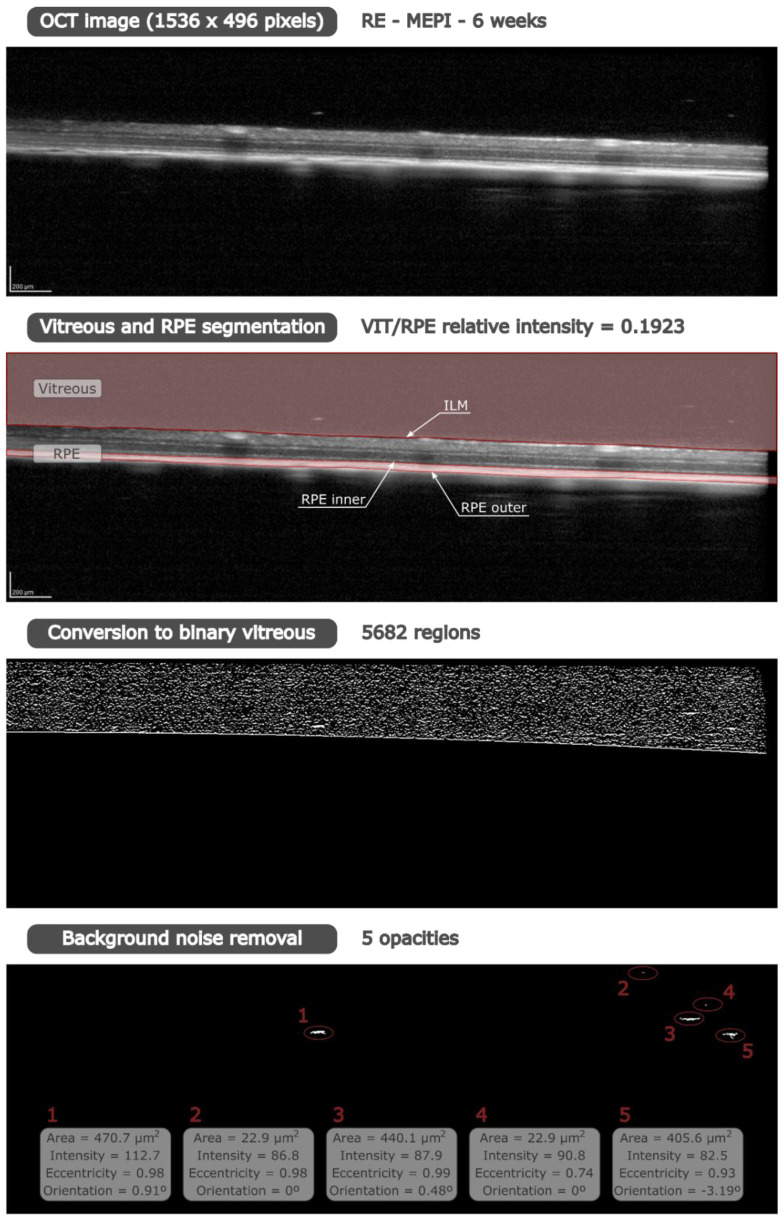
Methodology of the custom program implemented in Matlab (version R218a, Mathworks Inc., Natick, MA, USA). From an OCT (HR-OCT Spectralis, Heidelberg^®^ Engineering, Heidelberg, Germany) image of a right eye with the MEPI model at 6 weeks, VIT/RPE relative intensity was quantified after vitreous and RPE segmentation. To analyse the opacities/cells, it is necessary to remove the background noise of the vitreous. Our denoising filter was applied to the grey scale image intensity in those regions identified in the binary image. Therefore, intensity was obtained from the grey scale image, while area, eccentricity and orientation were computed from the binary image. Abbreviations: OCT: optical coherence tomography; RE: right eye; MEPI: glaucoma model induced by sclerosing the episcleral veins; VIT: vitreous; RPE: retinal pigment epithelium.

**Figure 2 biomedicines-09-01792-f002:**
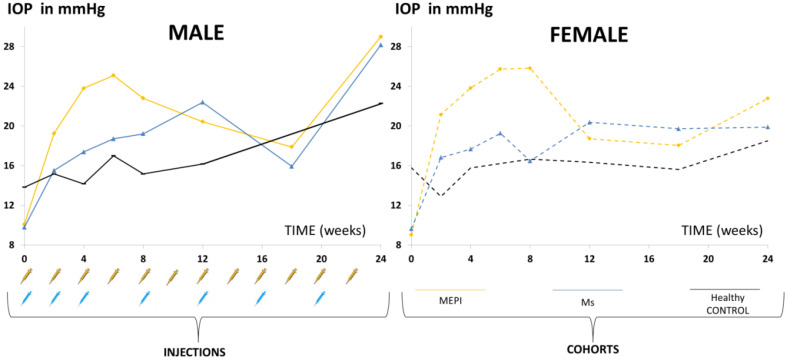
Intraocular pressure curves (right eyes) in two models of chronic glaucoma and healthy controls. Abbreviations: MEPI: glaucoma model induced by sclerosing the episcleral veins; Ms: glaucoma model induced by injection of PLGA microspheres; IOP: intraocular pressure (data extracted from [50,62]).

**Figure 3 biomedicines-09-01792-f003:**
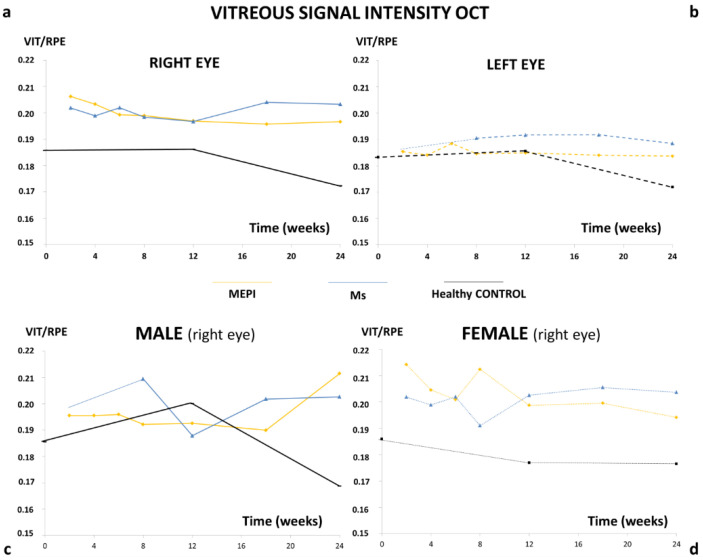
VIT/RPE signal intensity. (**a**) Right eye (both sexes); (**b**) left eye (both sexes); (**c**) males (right eye); (**d**) females (right eye). MEPI: episcleral vein sclerosis model (yellow); Ms: microsphere intraocular injection model (blue); healthy CONTROL: cohort of healthy animals without intervention (black); VIT: vitreous; RPE: retinal pigment epithelium.

**Figure 4 biomedicines-09-01792-f004:**
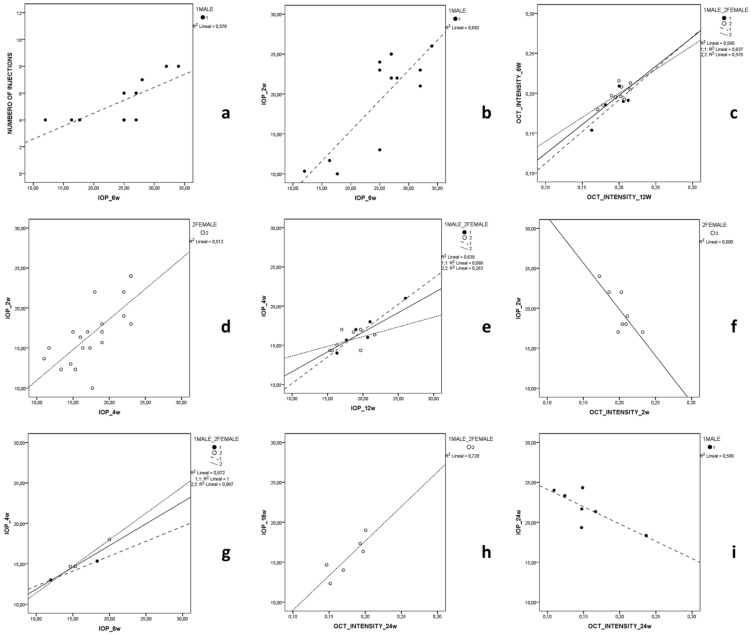
Significant correlations by sex in the two models of chronic glaucoma (episcleral model: (**a**–**c**); Ms model: (**d**–**f**)); and healthy controls (**g**–**i**). Abbreviations: IOP: intraocular pressure; OCT: optical coherence tomography; w: week.

**Figure 5 biomedicines-09-01792-f005:**
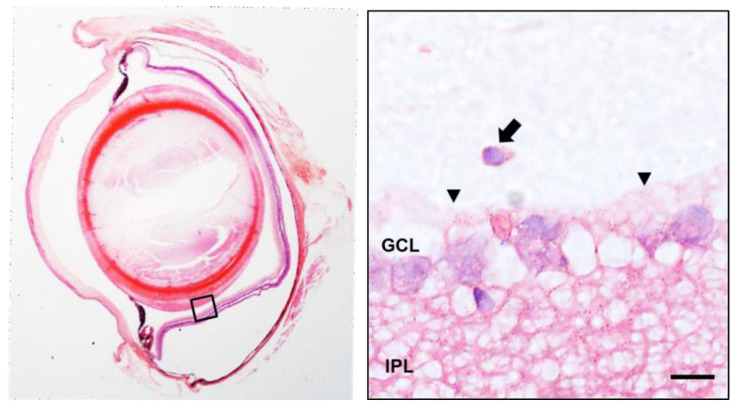
Localization of intravitreal cells (arrow) close to the internal limiting membrane (arrowhead) in hypertensive rat eyes. GCL: ganglion cell layer; IPL: internal plexiform layer. Scale bar: 12.5 µm.

**Figure 6 biomedicines-09-01792-f006:**
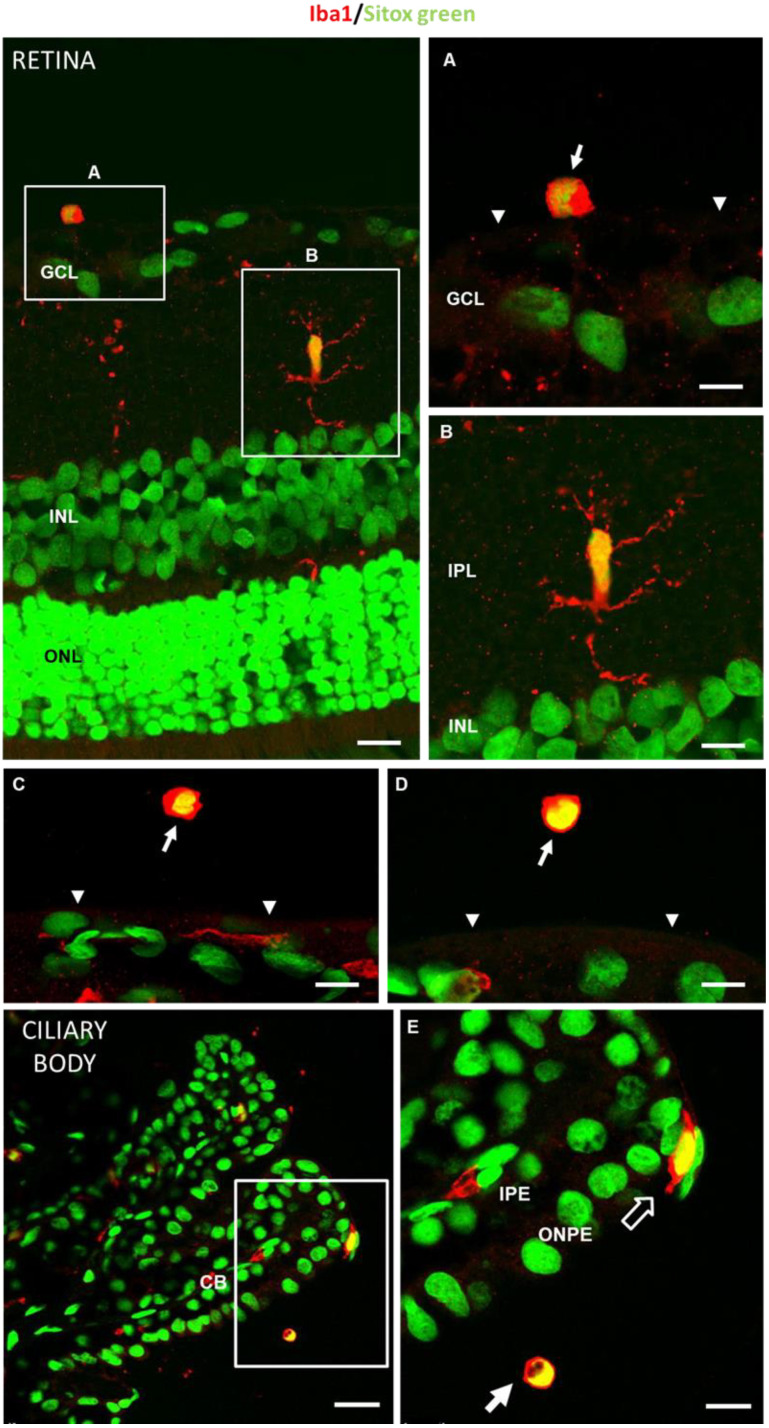
Hyalocyte-like cells in hypertensive rat eyes showed positive for Iba1 (arrows). (**A**,**C**,**D**,**E**): Examples of Iba1 positive hyalocyte-like cells. (**B**): Activated microglia embedded in the retina also showed, as expected, positive for Iba1. (**E**): The presence of Iba1 positive cells crossing the outer non-pigmented epithelium could suggest that hyalocytes migrate from the ciliary body to the vitreous. GCL: ganglion cell layer; IPL: internal plexiform layer; INL: inner nuclear layer; ONL: outer nuclear layer; arrowhead: internal limiting membrane; CB: ciliary body; ONPE: outer non-pigmented epithelium; IPE: inner pigmented epithelium; open arrow: Iba1 positive cell crossing the outer non-pigmented epithelium; white arrow: Iba1 positive hyalocyte. Scale bar: Retina: 12.19 µm; (**A**): 6.25 µm; (**B**): 7.24 µm; (**C**): 10 µm; (**D**): 8.3 µm; Ciliary body: 23.81 µm; (**E**): 9.92 µm.

**Figure 7 biomedicines-09-01792-f007:**
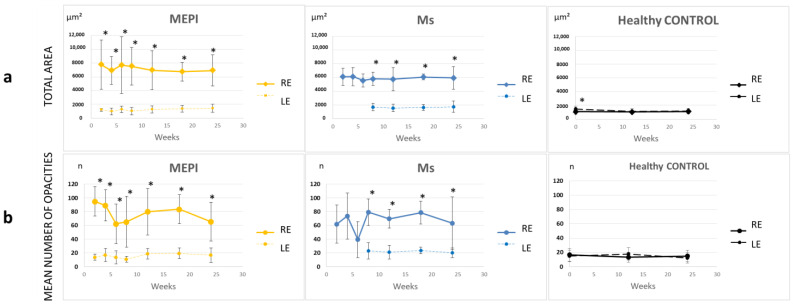
Changes in total immune response (**a**) and cellular quantification (**b**) in both glaucoma models and healthy controls. Abbreviations: RE: right eyes; LE: left eyes; MEPI: model induced by sclerosing the episcleral veins; Ms: model induced by injecting microspheres into the anterior chamber; n: number; *: statistical significance, *p* < 0.05, using ANOVA test.

**Figure 8 biomedicines-09-01792-f008:**
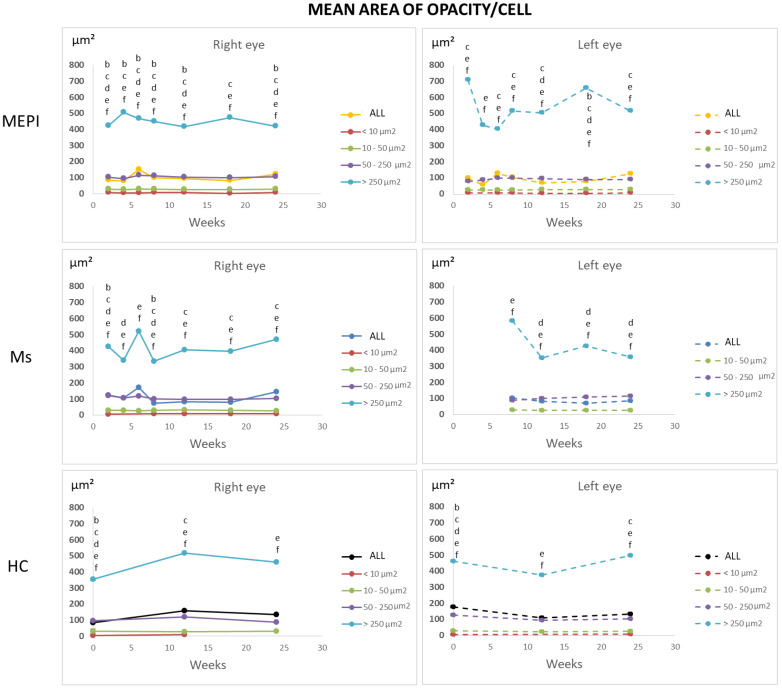
Cell subdivisions based on the mean area of vitreous opacities measured using OCT. Abbreviations: MEPI: model induced by sclerosing the episcleral veins; Ms: model induced by injecting microspheres into the anterior chamber; isolated cells: <10 microns^2^ (group 1); non-activated cells: 10–50 microns^2^ (group 2); activated cells: 50–250 microns^2^ (group 3); cell complexes: >250 microns^2^ (group 4). Cell populations maintain similar sizes over time, implying reliability of measurement. Complexes > 250 microns^2^ undergo the greatest variations with peaks at the onset of noxa in both eyes. Statistically significant differences (*p* < 0.05) were highlighted with alphabetic markers as follows: a (group 1-group 2); b (group 1-group 3); c (group 1-group 4); d (group 2-group 3); e (group 2-group 4); f (group 3-group 4).

**Figure 9 biomedicines-09-01792-f009:**
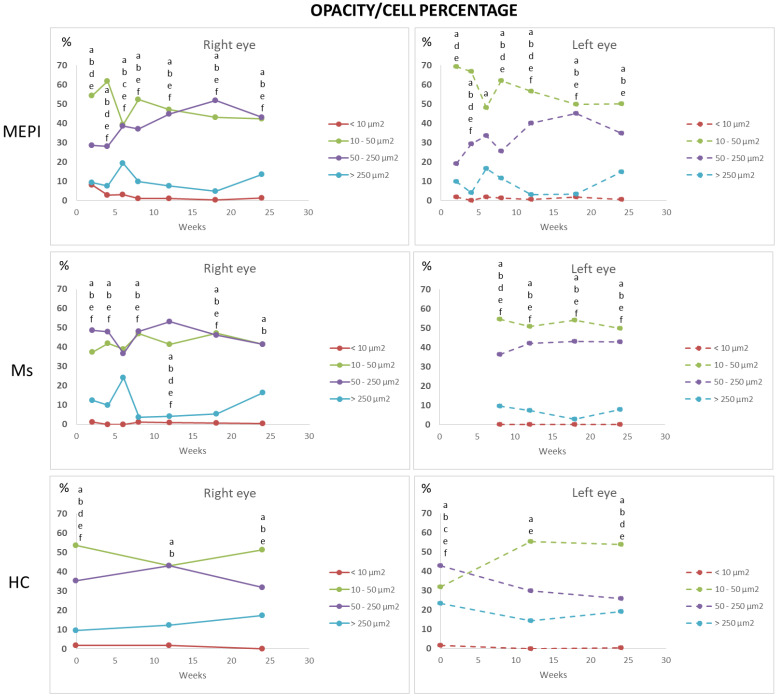
Changes in the vitreous immune population (opacities) in both chronic glaucoma models and healthy controls over 6 months. Abbreviations: MEPI: model induced by sclerosing the episcleral veins; Ms: model induced by injecting microspheres into the anterior chamber; isolated cells: opacities < 10 microns^2^ (group 1); non-activated cells: 10–50 microns^2^ (group 2); activated cells: 50–250 microns^2^ (group 3); cell complexes: > 250 microns^2^ (group 4). Data represented in percentages. Statistically significant differences (*p* < 0.05) were highlighted with alphabetic markers as follows: a (group 1-group 2); b (group 1-group 3); c (group 1-group 4); d (group 2-group 3); e (group 2-group 4); f (group 3-group 4).

**Figure 10 biomedicines-09-01792-f010:**
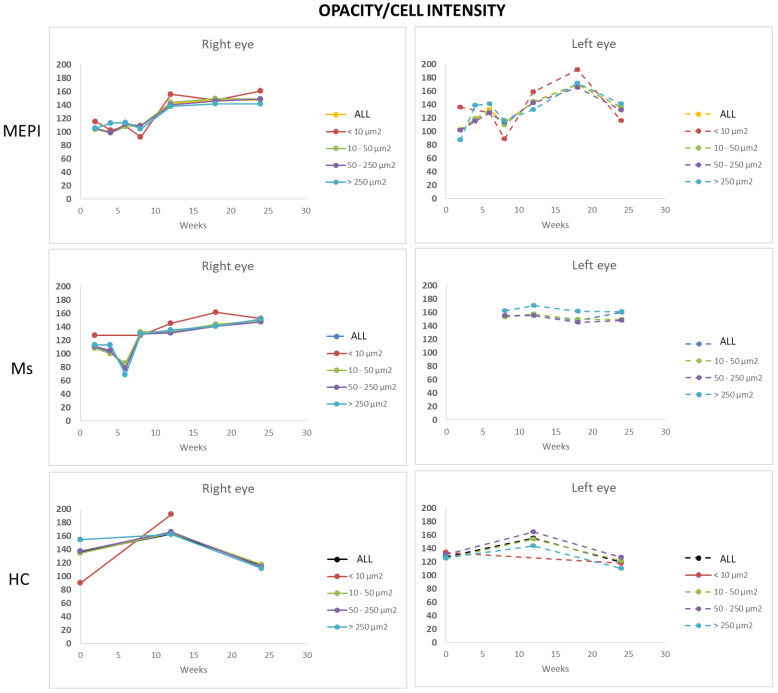
Mean intensity of opacities/cells based on size in both chronic glaucoma models and healthy controls. Abbreviations: MEPI: model induced by sclerosing the episcleral veins; Ms: model induced by injecting microspheres into the anterior chamber; isolated cells: opacities < 10 microns^2^; non-activated cells: 10–50 microns^2^; activated cells: 50–250 microns^2^; cell complexes: > 250 microns^2^. There were no statistically significant differences.

**Figure 11 biomedicines-09-01792-f011:**
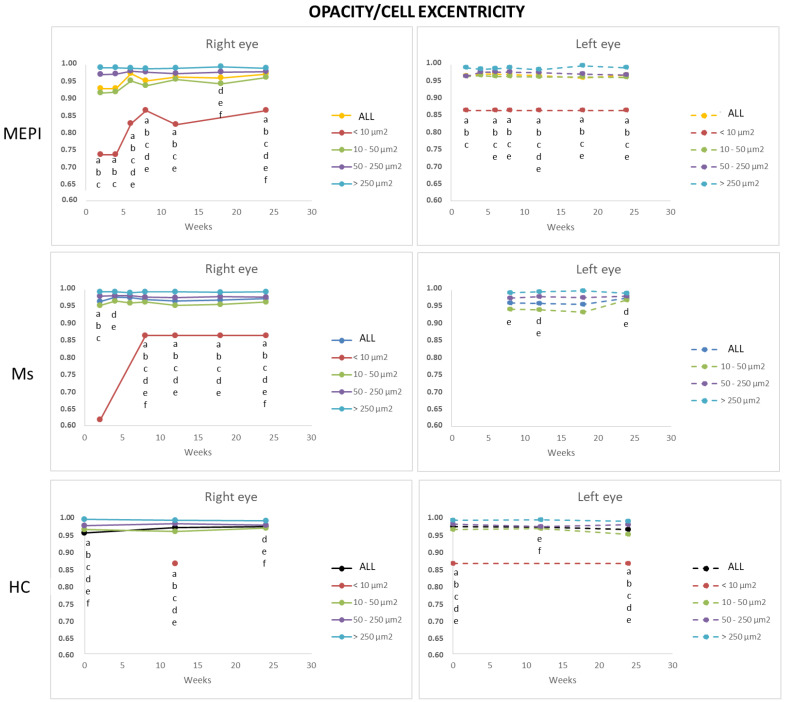
Mean eccentricity of vitreous opacity detected using OCT, according to size, in both glaucoma models and healthy controls. Indirect in vivo analysis of cell soma morphology. Abbreviations: MEPI: model induced by sclerosing the episcleral veins; Ms: model induced by injecting microspheres into the anterior chamber; isolated cells: opacities < 10 microns^2^ (group 1); non-activated cells: 10–50 microns^2^ (group 2); activated cells: 50–250 microns^2^ (group 3); cell complexes: > 250 microns^2^ (group 4). Statistically significant differences (*p* < 0.05) were highlighted with alphabetic markers as follows: a (group 1–group 2); b (group 1–group 3); c (group 1–group 4); d (group 2–group 3); e (group 2–group 4); f (group 3–group 4).

**Figure 12 biomedicines-09-01792-f012:**
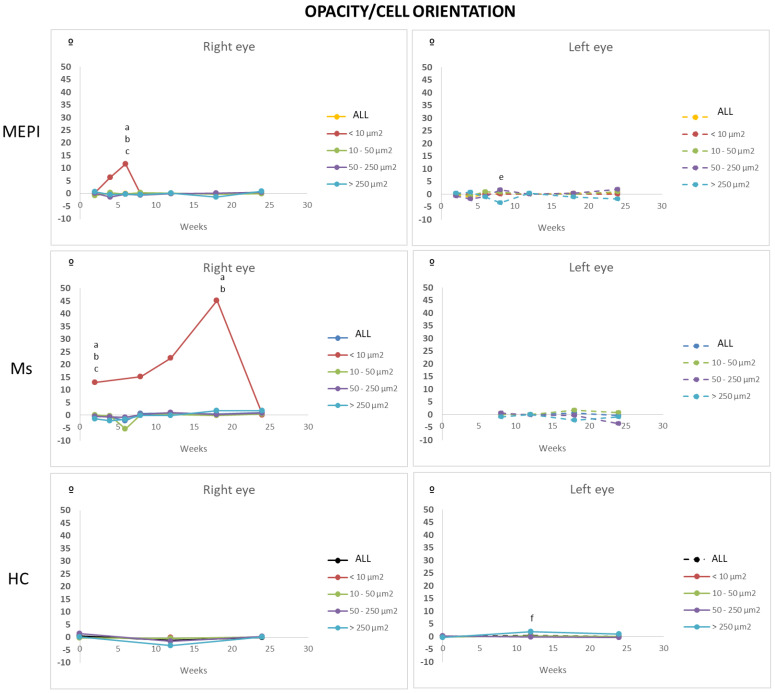
Mean orientation of vitreous opacity detected using OCT, according to size, in both glaucoma models and healthy controls. In vivo analysis of the change in opacity/cell direction for motility. Abbreviations: MEPI: model induced by sclerosing the episcleral veins; Ms: model induced by injecting microspheres into the anterior chamber; isolated cells: opacities < 10 microns^2^ (group 1); non-activated cells: 10–50 microns^2^ (group 2); activated cells: 50–250 microns^2^ (group 3); cell complexes: > 250 microns^2^ (group 4). Statistically significant differences (*p* < 0.05) were highlighted with alphabetic markers as follows: a (group 1–group 2); b (group 1–group 3); c (group 1–group 4); d (group 2–group 3); e (group 2–group 4); f (group 3–group 4).

**Table 1 biomedicines-09-01792-t001:** Correlations in both chronic glaucoma models and healthy controls. Abbreviations: INJ: injections; IOP: intraocular pressure; OCT: optical coherence tomography; w: week; RE: right eyes; LE: left eyes; MEPI: model induced by sclerosing the episcleral veins; Ms: model induced by injecting microspheres into the anterior chamber; HC: healthy controls; im: inverse moderate correlation; m: moderate correlation. In bold: statistically high correlations.

	RIGHT EYE			LEFT EYE		
	MEPI	Ms	HC	MEPI	Ms	HC
**INJ/IOP**	Inj/6 w (m)	Inj/8 w (m)				
**IOP/IOP**	0 w/8 w (im) 2 w/4–6–8 w (m) **2 w/24 w** (r = 0.816, *p* = 0.025)	2 w/4 w (m) 8 w/12 w (m) **8 w/24 w** (r = 0.917, *p* = 0.028)	**4 w/8 w** (r = 0.934, *p* = 0.020)	0 w/8 w (im)	**4 w/12 w** (r = 0.800, *p* = 0.004)	
**IOP/OCT**	**2 w/2 w** (r = 0.812, *p* = 0.027)			**0 w/18 w** (r = −0.979 *p* = 0.004) **24 w/24 w** (r = 0.885, *p* = 0.008)	**18 w/18 w** (r = 0.890 *p* = 0.043)	
**OCT/OCT**	6 w/12 w (m) (r = 0.762, *p* = 0.001)					

## Data Availability

On request to corresponding author.

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
