# Peer review of "Analysis of Parainflammation in Chronic Glaucoma Using Vitreous-OCT Imaging"

_biomedicines, 2021, doi:10.3390/biomedicines9121792_

Round 1
Reviewer 1 Report
The authors present a study that monitors vitreous parainflammation in two animal models of glaucoma, using vitreous hyperreflective opacities on OCT images, as indicators of immune response. The study is based on the computational analysis of such opacities, for obtaining metrics ay different stages of the follow-up. The topic of the study is very relevant, and the findings may have clinical impact. However, there are methodological issues, mainly related to image processing and statistical analysis, that must be clarified.
1 – What modifications were done to the optics of the Spectralis system to allow imaging of the rat’s retina? The authors mention the use of a “plane power 145 polymethylmethacrylate contact lens (…) adapted to the rats’ cornea”. However, they do not provide the power of the lens and the expression “plane power” is confusing. This lens modifies the vergence of the light beam or just increases the optical path on the sample arm of the OCT interferometer?
2- It is stated that the image area is 2.906 mm2. The authors should explain how they measured this area since, due to the optics of the rat’s eye, this field of view is obviously different from that obtained when using the Spectralis on humans.
3 – During image processing, the greyscale OCT images were converted to binary images. How was defined the binarization threshold? Was it adaptive or fixed for all images? How was controlled the gain during OCT image acquisition?
4 – The authors mention the use of a denoise filter after binarization, to remove speckle noise. The text does not include the type and characteristics of the used filter and unfortunately, due to a reference error (line 194) its is not possible to identify the reference where this information is provided. Without that it is impossible to verify if the used filter is adequate. This is import since many de-speckling filters impact on the interfaces of image structures changing metrics like area of objects.
5 – The authors used ANOVA multiple comparisons tests to assess differences between the mean values of the variables under study listed on section 2.4. However, the results of these tests are never mentioned in the manuscript. As the plots of figures 6 – 10 do not include error bars, it is not possible to evaluate if statistically significant differences were found.
6 – ANOVA multiple comparisons tests only detects if statistically significant differences are present. They do not identify where those differences lie. A post-hoc test (like the Tukey test) should be used for that purpose. In situations when there are a series of observations over time, the Bonferroni method is not appropriate, as it will be highly conservative and may miss real differences.
7 – The use of scatter plots (figures 6 – 10) connected with soft curves is not recommended since it may imply the existence of an analytical function that describes the data. If the intension is to show how the variables evolved, then connect the data points with straight lines.
Formatting issues
Line 47: reference to AGIS is not numerical.
Lines 164 and 193: reference error (Error! Reference source not found).
References 23 and 75: the paper title is written in upper case.
Reference 69: several errors in the names of the authors.
Reviewer 2 Report
THE AUTHORS examined OCT findings and parainflammatory cells (microglial cells) in rat glaucoma models. This study is interesting and well-organized. I have concerns about the manuscript as follows:
- The terms “inner limiting membrane” should be changed to internal limiting membrane throughout the manuscript.
- In line 164 and 193, the description of “Error! Reference source not” and citation of the references should be corrected
- In figure 5C, what are cells located near arrowheads indicating ILM in the figure? In line 353, the authors stated that “However, this cannot be ruled out”. What do you mean? Please explain more.
- In figure 6, the labels of a, b, and c are missing in each figure.
- In line 309, the term HTELTHY should be changed to Healthy.
- I feel that all the figures and tables are needed in the article, so supplementary files are not needed.
Round 2
Reviewer 1 Report
The authors did a very good job correcting the manuscript and provided adequate replies and corrections to my review. The paper is now ready to be published except for one point.
From the text on section 2.2 (page 6) and the caption of figure 1 it is possible to think get the idea that the denoising filter was applied on the binarized image. That, obviously, cannot be true since, on the binarized image, there are only two possible pixel values (0 and 1), and the filter is based on the intensity of the areas. My interpretation is that the binarized image was used to identify the regions to be analyzed in the vitreous (i.e., like a mask) and the computation of the intensities was performed on the gray scale image. However, I am not sure if my interpretation is correct. Anyway, the text should be clear enough to allow an easy understanding of the image processing steps and their sequence.
Author Response
The authors did a very good job correcting the manuscript and provided adequate replies and corrections to my review. The paper is now ready to be published except for one point.
From the text on section 2.2 (page 6) and the caption of figure 1 it is possible to think get the idea that the denoising filter was applied on the binarized image. That, obviously, cannot be true since, on the binarized image, there are only two possible pixel values (0 and 1), and the filter is based on the intensity of the areas. My interpretation is that the binarized image was used to identify the regions to be analyzed in the vitreous (i.e., like a mask) and the computation of the intensities was performed on the gray scale image. However, I am not sure if my interpretation is correct. Anyway, the text should be clear enough to allow an easy understanding of the image processing steps and their sequence.
Author’s response: The authors thank reviewer for this comment. The reviewer’s interpretation is totally correct, but it is true that this point needs to be well explained in the manuscript. We have clarified, both in section 2.2 and in the caption of figure 1, that our denoising filter was applied to the grey scale image using binary image to detect all regions in the vitreous. Thus, opacity intensity was obtained from the grey scale image and the rest of parameters (number, area, eccentricity and orientation) were calculated from binary image.
Caption of figure 1: “Our denoising filter was applied to the grey scale image intensity in those regions identified in the binary image. Therefore, intensity was obtained from the grey scale image, while area, eccentricity and orientation were computed from the binary image.”
Section 2.2: “In this way, opacity intensity was obtained from grey scale image using binary image as a mask to detect the regions. Moreover, number, size, eccentricity and orientation of these opacities were obtained from binary image.”